# A Preliminary Study Indicating Improvement in the Median Survival Time of Glioblastoma Multiforme Patients by the Application of Deuterium Depletion in Combination with Conventional Therapy

**DOI:** 10.3390/biomedicines11071989

**Published:** 2023-07-13

**Authors:** Gábor Somlyai, Beáta Zsuzsanna Kovács, András Papp, Ildikó Somlyai

**Affiliations:** 1HYD LLC for Cancer Research and Drug Development, Villányi út 97, 1118 Budapest, Hungary; bzskovacs@hyd.hu (B.Z.K.); isomlyai@hyd.hu (I.S.); 2Department of Public Health, Albert Szent-Györgyi Medical School, University of Szeged, 6720 Szeged, Hungary; papp.andras@med.u-szeged.hu

**Keywords:** deuterium depletion, deuterium-depleted water (DDW), glioblastoma multiforme, median survival time (MST)

## Abstract

Glioblastoma multiforme (GBM) and malignant gliomas are the most common primary malignant brain tumors. Temozolomide (TMZ) chemotherapy plus radiation therapy (RT), admi-mistered after debulking surgery, increased the median survival time (MST) from 12.1 months with RT alone merely to 14.6 months, respectively. In this study, the actions of deuterium-depleted water (DDW) on the survival of GBM patients who also received conventional therapies was investigated. Without changing the conventional treatment, the daily fluid intake of the patients was wholly replaced with DDW in 1.5–2 L per day volume to reduce the D concentration in their bodies. The primary endpoint was the MST. The 55 patients involved in this study, who received conventional treatment and consumed DDW, showed a longer MST (30 months) compared to the historical control (12.1–14.6 months). There was a massive difference between the two genders in the calculated MST values; it was 25 months in the male subgroup (n = 33) and 42 months in the female subgroup (n = 22), respectively. The MST was 27 months without TMZ treatment (38 patients) and 42 months in the TMZ-treated group (17 patients), respectively. For the selected 31 patients, who consumed DDW in the correct way in addition to their conventional treatments, their MST was calculated as 30 months. Within this group, the 20 subjects who had relapsed before DDW treatment had 30 months of MST, but in those 10 subjects who were in remission when DDW treatment started, their MST was 47 months. In the subgroup of patients who began their DDW treatment parallel with radiotherapy, their MST was again 47 months, and it was 25 months when their DDW treatment was started at 8 weeks or later after the completion of radiotherapy. Altogether, these survival times were substantially prolonged compared to the prospective clinical data of patients with primary GBM. Consequently, if conventional therapies are supplemented with D depletion, better survival can be achieved in the advanced stage of GBM than with the known targeted or combination therapies. Application of DDW is recommended in all stages of the disease before surgery and in parallel with radiotherapy, and repeated DDW courses are advised when remission has been achieved.

## 1. Introduction

Primary brain tumors are a heterogeneous group of benign and malignant neoplasms, arising from the brain parenchyma and its surrounding structures. These tumors are an important cause of morbidity and mortality in both adults and children [1,2]. The overall pooled incidence rate of primary brain tumors was found to be 10.82 per 100,000 person-years [3]. Glioblastoma multiforme (GBM) and malignant gliomas are the most common primary malignant brain tumors reported in the USA, with an annual incidence of 5.26 per 100,000 population or 17,000 new diagnoses per year and are typically associated with a dismal prognosis and a poor quality of life [4]. In glioblastoma multiforme, defined as an aggressive primary brain tumor, the usual therapies are barely efficient. Even the most complex approach, debulking surgery followed by irradiation plus temozolomide (TMZ) chemotherapy, merely increased the median survival time to 14.6 months vs. 12.1 months with radiation therapy (RT) alone. Moreover, progression is delayed by RT + TMZ by only 6.9 months [5]. In recurrent gliomas, the progression-free survival and the response to TMZ are much worse [6].

Clinical complications (thromboembolic events, seizures, and fluctuation of existing neurologic symptoms) are frequently observed in patients suffering from malignant gliomas. Unwanted effects due to corticosteroids and chemotherapeutic agents are also common and need to be managed properly. GBM, despite the improving therapeutic options, is still very difficult to treat. Optimal management requires a multidisciplinary approach and knowledge of the potential complications from both the disease and its treatment.

The possible role of naturally occurring deuterium (16.8 mmol/L in natural waters) was investigated in different studies, which revealed that deuterium depletion inhibited tumor cell growth in vitro and caused tumor regression in vivo [7,8,9,10,11,12]—apparently due to the significantly different chemical and physical properties of hydrogen (H) and deuterium (D) [13,14], and because changing the D/H ratio exerts a significant impact on cell physiology [15,16]. Numerous studies conducted on different cell lines in culture media containing deuterium-depleted water (DDW) verified the determinative role of D in cell cycle regulation and tumor growth [12,17,18,19,20]. D depletion influences protooncogenes and tumor suppressor genes. When experimental animals were given DDW to drink, the induction of the expression of the c-Myc, Ha-ras, and p53 genes by carcinogen exposure was significantly inhibited [7]. Further, the apoptosis-triggering effect of DDW has been demonstrated both in vitro [17] and in vivo [21]. D, as a natural cell growth regulator, also controls the mitochondrial oxidation–reduction balance [22,23,24]. D depletion causes a disbalance between the production and neutralization of reactive oxygen species (ROS) in the mitochondria, inducing oxidative stress, which in turn induces apoptosis [22].

The clinical outcome of D depletion was investigated in prospective, phase 2, double blind [10], and retrospective human clinical studies [10,11,25,26]. For retrospective studies, clinical data on the test results of the conventional therapies in patients consuming DDW were collected. Patients with early-stage breast cancer (n = 158) achieved a median survival time (MST) of 217 months (18.1 years), and those with advanced disease (n = 74) obtained a median survival time of 52 months (4.3 years), compared to 17 months in patients not consuming DDW, respectively. Patients who took a single course of DDW (n = 126) had an MST of 9 years, whereas patients (n = 53) who took at least two courses of DDW attained an MST of 24.4 years, respectively [25]. In another study on patients with small cell and non-small cell lung cancers who consumed DDW, the MST was 25.9 months in male and 74.1 months in female patients, respectively—2–4 times longer than it has been generally observed in lung cancer patients [11,26].

It is known that median survival is longer in GBM patients who express the IDH1 (R132H) mutation compared to patients who express the wild allele [27,28]. IDH1-derived 2-hydroxyglutarate can facilitate the degradation of Hif1-α and thus reduce the Warburg effect through the downregulation of multiple genes in the glycolytic pathway [29]. Further evidence of an inhibitory effect of the IDH1 mutation on glucose intake and glycolysis was recently obtained from PET analysis [30].

A ketogenic diet has been used as complementary cancer therapy with high efficacy [31]. A synergistic interaction between the effects of the IDH1 mutation and ketogenic metabolic therapy (KMT) could simultaneously downregulate both the Warburg effect and the Q-effect [32] in GBM neoplastic cell populations, thus providing a novel mechanism contributing to the long-term survival of the patients [33]. Ketogenic diets can facilitate the delivery of small-molecule therapeutic drugs through the blood–brain barrier without toxicity. As GBM, like most malignant tumors, is dependent on fermentation for ATP synthesis and survival, the simultaneous restriction of fermentable fuels, glucose, and glutamine, while elevating the non-fermentable ketone bodies, offers a non-toxic therapeutic strategy for managing GBM [34,35,36].

Based on the proven anticancer effects of DDW, we postulate that the beneficial impact of the ketogenic diet in cancer treatment derives at least partly from its deuterium-depleting effect, since the mitochondria, when oxidizing fats instead of carbohydrates, produce metabolic water with as low as a 118 ppm D level due to the differences in the D content of the various nutrients [37].

In this paper, we present a retrospective study based on hospital records of 55 GBM patients between 1994 and 2020, respectively. Our aim was to investigate how DDW consumption influences the outcome of GBM in combination with the conventional therapies. These were withheld, but the patients’ daily water demand was fully covered with DDW. The primary endpoint was the MST.

The data indicated that D depletion, when combined with the existing conventional therapies, provides a several-fold increase in the MST.

## 2. Materials and Methods

### 2.1. Study Population

Patients suffering from GBM and receiving conventional forms of therapy plus follow-up examinations in different hospitals throughout Hungary were involved in this retrospective study. Altogether, 55 GBM patients started consuming DDW between April 1994 and October 2020, respectively. Data collection was closed in January 2021. Some of the patients drank DDW for a far shortened period (i.e., less than 4 months) or in an inappropriate way (e.g., alternating DDW with normal fluids, or drinking DDW of a insufficient D concentration). After excluding these subjects from the study, 33 patients remained for detailed evaluation. However, 2 of them started drinking DDW years after the diagnosis, meaning they had to be omitted not to cause an erroneous MST. Ultimately, 21 males and 10 females were enrolled in the detailed retrospective evaluation, respectively.

### 2.2. Administration of DDW

The technology of DDW and the deuterium-depleted drinking water (Preventa) production for human consumption was detailed earlier [11]. During the treatment, the patients covered their daily water demand by drinking 1.5 to 2 L of Preventa DDW—with 85, 65, 45, and 25 ppm D concentration, respectively—instead of normal water, while their conventional therapy was also continued. Due to the poor prognosis of GBM, DDW consumption in our cases typically started at the 85 ppm D level, and was changed to 65, 45, and 25 ppm after every 1-to-3 months in order to maintain a continuous depletion of deuterium. In cases where a longer progression-free interval was achieved, a 3-to-6 months break in DDW consumption was introduced, followed by further DDW course of 4-to-6 months lengths. Following this, DDW application was shortened to 3–4 months, and the breaks were prolonged accordingly. The overall duration of DDW consumption in the whole follow-up population was between 22 and 1566 days, respectively.

The patients were continuously updated with, aware of, consented, and provided access to all the available information and publications regarding deuterium depletion.

### 2.3. Statistical Evaluation

The sole endpoint of the study was survival. Kaplan–Meier survival estimates were used along with the log-rank test to compare groups. All statistical computations were performed using SPSS v25 software (IBM Corp., Armonk, NY, USA). The study was performed retrospectively, and all statistical results were declared statistically significant with *p* < 0.05.

For correlation analysis, the Pearson’s correlation coefficient was used. The calculations were performed by Adware Research Ltd. (Balatonfüred, Hungary).

## 3. Results

### 3.1. Evaluation of the Whole Study Population of 55 GBM Patients Consuming DDW

#### 3.1.1. Characteristics of the Whole Study Population

All patients involved in the analysis had a positive diagnosis of GBM before starting DDW application. The tumor was present in the beginning in all but two subjects who were tumor free thanks to a previous operation. All of the received conventional therapies, including chemotherapeutical, irradiation, surgery, targeted therapy, or a combination of these conventional therapies. The time between the diagnosis and beginning of DDW consumption was varied.

#### 3.1.2. Median Survival Time (MST) of the Whole Study Population

The 55 patients consuming DDW, including even those for whom DDW consumption lasted only for 22–123 days, showed a longer MST (95% CI: 9.4–50.5) of 30 months compared to historical control (12.1–14.4 months) (Figure 1).

#### 3.1.3. Differences of the MSTs in the Whole Cohort by Gender

There was a massive (but not significant, *p* = 0.283) difference between the two genders in the calculated MST values. The median survival time was 25 months (95% CI: 15.9–34.0 months) in the male (33 patients) group, and 42 months (95% CI: 18.3–65.6 months) in the female (22 patients) subgroup, respectively (Figure 2).

#### 3.1.4. Correlation between the Survival Times and the Duration of DDW Consumption in the Whole Study Population

Pearson’s correlation coefficient (r) was calculated to assess to what extent the length of using DDW and the survival time taken from the start of DDW consumption are correlated. An “r” value of 0.692 was obtained (well above 0.5, from which a strong correlation is assumed), indicating that longer DDW consumptions result in extended survival times (Figure 3). The dots above the slope show that a shorter duration of DDW consumption can also result in longer survival but may also indicate heterogeneity of the evaluated population regarding the staging, size of the tumor, and the conventional treatment received.

#### 3.1.5. MST in the Whole Study Population Stratified by Temozolomide Treatment

The majority of the patients had been recruited years before temozolomide (TMZ) was registered, and access to TMZ remained limited for several years after its registration. This gave an opportunity to evaluate survival in a TMZ-naïve and a TMZ-treated subpopulation. The MST achieved in the former group (38 patients) was 27 months (95% CI: 18.8–35.1 months), and in the latter (17 patients) 42 months (95% CI: 14.6–69.3 months), respectively, with no statistically significant differences observed between these two groups (*p* = 0.797). However, the MST of both subpopulations was about three-fold of the historical control (Table 1).

### 3.2. Detailed Evaluation of the Selected 31 GBM Patients

The patients who started consuming DDW at different times (0–542 days, median: 57 days) after their diagnosis and the length of DDW consumption varied between 133 and 1566 days, respectively.

#### 3.2.1. Median Survival Time of the Selected 31 GBM Patients

MST of the selected 31 patients who received DDW treatment along with the conventional therapies was substantially longer vs. the historical control. Their median survival time was 30 months (95% CI: 0.5–59.4) (Figure 4), suggesting that this subset of cases was representative for the whole cohort of 55 patients.

#### 3.2.2. Calculation of the MST in the Selected 31 GBM Patients Stratified by Relapse Status

These 31 cases were stratified into two categories: being in remission or having had relapse before commencing DDW consumption.

The MST was 30 months in the subgroup of patients (20 subjects) who had a relapse before DDW treatment, and 47 months in the subgroup of patients (10 subjects) who were in remission when their DDW treatment started (and, accordingly, relapsed only later) (Table 1), respectively. However, due to their small sample size, this difference was not determined to be significant (*p* = 0.246). One patient who had no relapse was excluded from this analysis.

#### 3.2.3. Calculation of the MST in the Study Population Stratified by Radiotherapy

The subjects were stratified on the basis whether DDW treatment started together with radiotherapy (simultaneously or within 8 weeks after last radiation) or more than 8 weeks after radiotherapy was completed. The MST was calculated as 47 months (95% CI: 18.8–75.1) in the subgroup of patients who started DDW treatment together with radiotherapy, and in the other group it was 25 months (95% CI: 14.5–35.4) (Figure 5), respectively. However, the sample size was too small to detect significant differences (*p* = 0.942).

## 4. Discussion

The two isotopes of hydrogen; protium (^1^H) and deuterium (^2^H) have a 100% mass difference, resulting in significant alterations in both their chemical and physical properties [13,14]. The effect of D at elevated concentrations has been intensively investigated, [15,16,38] but these studies ignored the presence of naturally occurring D at a 16.8 mmol/L concentration.

It has previously been demonstrated through both in vitro and in vivo experiments that naturally occurring deuterium is a core factor in cell growth, and deuterium depletion induces apoptosis in tumor cells, resulting in a partial or complete tumor regression [9,12,17,21].

The subjects involved in the study were dissimilar in their stage of disease when their use of DDW started, in the delay between the diagnosis and start of DDW consumption, and what conventional therapies they received (a typical situation that arises when data are collected and analyzed retrospectively). To extract the maximum information on how efficient DDW is when applied in combination with conventional therapies, first the survival of all the 55 GBM patients involved was evaluated, following which (after the exclusion of those with a too short DDW consumption period or long survival without DDW) 31 of them were analyzed in detail. All results were summarized in Table 1.

The duration of DDW consumption and the length of survival showed a positive correlation (r = 0.692), which further supported that DDW in GBM patients indeed exhibited antitumor effect (Figure 3).

Survival times observed in the present study were markedly longer compared to the prospective clinical data of patients with primary GBM [3,4,5,6]; suggesting that D depletion and conventional treatments together can provide a longer survival in advanced GBM than any other targeted or combination therapy to date.

In the light of the scientific evidence on the biological effects of deuterium depletion [9,12,18,22,39], an important aim was to find the best combination of DDW and conventional therapies. The data in Table 1 show that DDW consumption enhanced the efficacy of both TMZ and radiotherapy. The 47 months’ MST of patients being in remission at the start of DDW consumption is in line with our earlier data showing that DDW increased progression-free interval and/or prevented relapse [7,10,11,25]. One recommended protocol option would be to start DDW consumption after operation, continue it during radiotherapy by consuming DDW with an 85 ppm D concentration, and reduce the D concentration to 65 ppm 2–3 weeks after the last radiation, combining with TMZ treatment applied according to the Stupp protocol [5], and after 1–3 months reduce the D concentration to 45 ppm and 25 ppm.

The substantial difference between the MST of females (42.0 months) and males (25.0 months) may be due to the different expression of oncogenes in response to deuterium depletion, as it had been previously observed in female and male mice [7]. Moreover, in another human study on non-small cell lung cancer, a striking difference was found between the calculated MSTs (males, 41.2 months; and females, 107.0 months; taken from the date of diagnosis) [11].

Oral DDW treatment has been proven to be safe and innocuous, according to preclinical toxicology studies [40], as well as from prospective and retrospective clinical trials [8,10,25,26].

The in vitro growth inhibition of tumorous cell lines under deuterium-depleted conditions, the clinically observed prolongation of progression-free interval, and the prevention of relapse in breast and lung cancer patients [25,26] were found to be in agreement. Consuming 1.5–2 L of DDW with 105 ppm D content day by day resulted in an inner D level decrease of about 1 ppm per day. Prolonged use of DDW with the same D level led to equilibrium. For the ongoing reductions of D concentration in the body fluids, a change to DDW of 20 ppm less D content is advisable every 2–3 months. The massive effects of a subnatural D concentration on a cellular level, including the induction of apoptosis [9,12,17,20,41] along with the inhibition of tumor cell migration [11], may both be of importance in terms of MST lengthening and relapse prevention observed in tumorous patients ingesting DDW [8,10].

The changes in the metabolic parameters of patients undergoing a ketogenic diet along with their beneficial antitumor effects have been well-documented [42,43,44]. The naturally low-D lipids in a ketogenic diet exhibit a significant inhibitory effect on tumor growth by preventing the cells from raising the D/H ratio to the threshold.

We conclude that D-depletion offers additional benefits in addition to conventional therapies and can be integrated into standard treatment regimens for GBM. The best option for GBM patients is achieving improvements by combining DDW with conventional therapies. The application of DDW is recommended in all phases of the disease, before surgery, during radiotherapy, and in repeated DDW courses when remission has been achieved.

This retrospective study has its obvious limitations but may help with the assessment of the necessity and feasibility of prospective studies.

## 5. Conclusions

Deuterium depletion substantially prolonged the survival time of GBM patients in an advanced stage. We conclude that D depletion offers additional benefits to conventional therapies and has a significant role in delaying progression. Application of DDW is recommended in all stages of the disease before surgery, parallel to radiotherapy and chemotherapy.

## Figures and Tables

**Figure 1 biomedicines-11-01989-f001:**
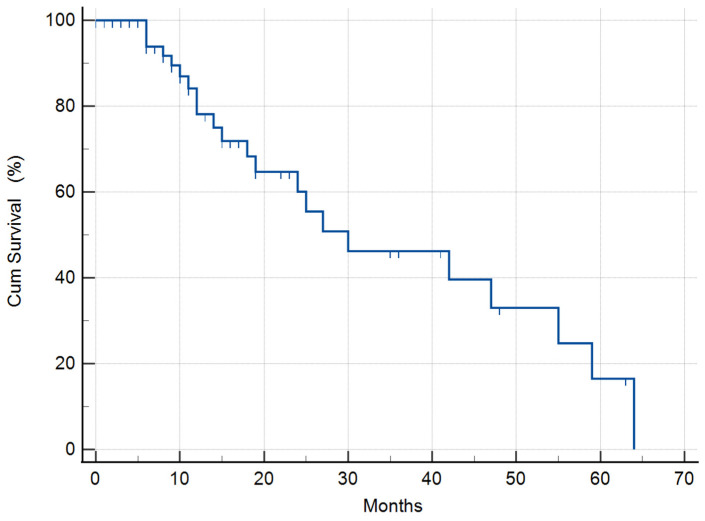
Kaplan–Meier graphs on the survival of the whole study population of 55 GMB patients (33 males, 22 females), showing a 30 (95% CI: 9.4–50.5) months MST, compared to the 12.1–14.4 months MST based on the historical data of GBM patients. The hash marks on the survival curve indicate when a patient was censored; the patient’s survival time exists.

**Figure 2 biomedicines-11-01989-f002:**
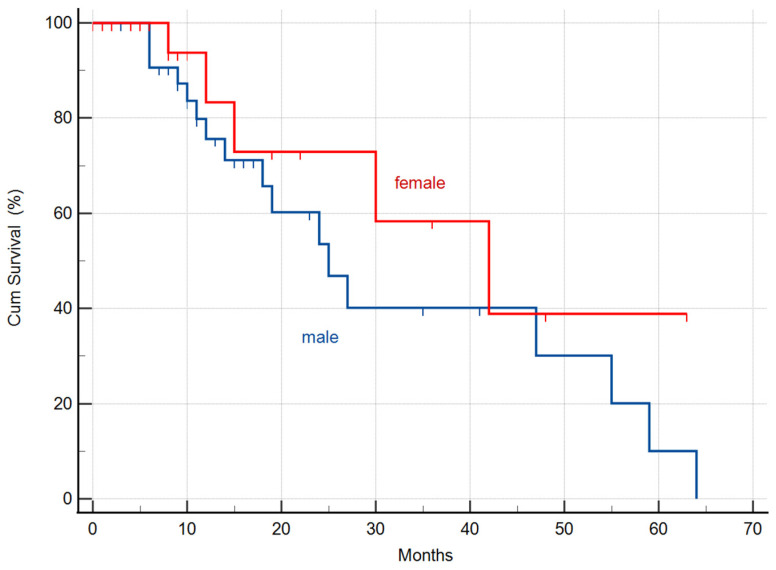
Kaplan-Meier graphs of survival in the male (33) and female (22) subgroups showed a 25 months (95% CI: 15.9–34.0) and 42 months (95% CI: 18.3–65.6) MST, respectively.

**Figure 3 biomedicines-11-01989-f003:**
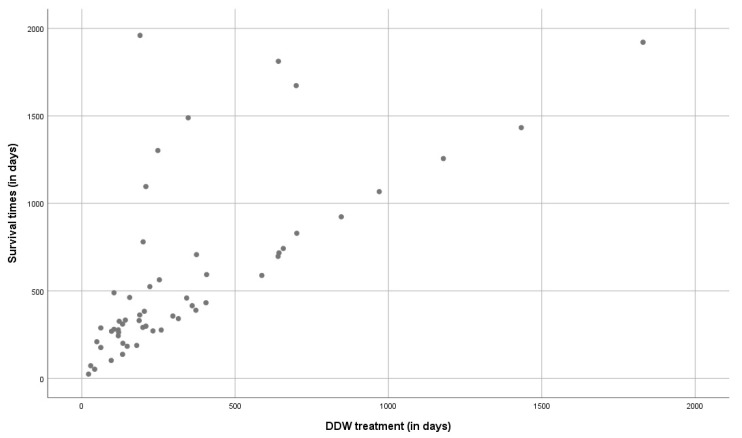
Correlation between the survival times and the duration of DDW consumption in the whole study population of 55 GBM patients. The “r” value of 0.692 was obtained, indicating a strong correlation between the length of DDW consumption and survival.

**Figure 4 biomedicines-11-01989-f004:**
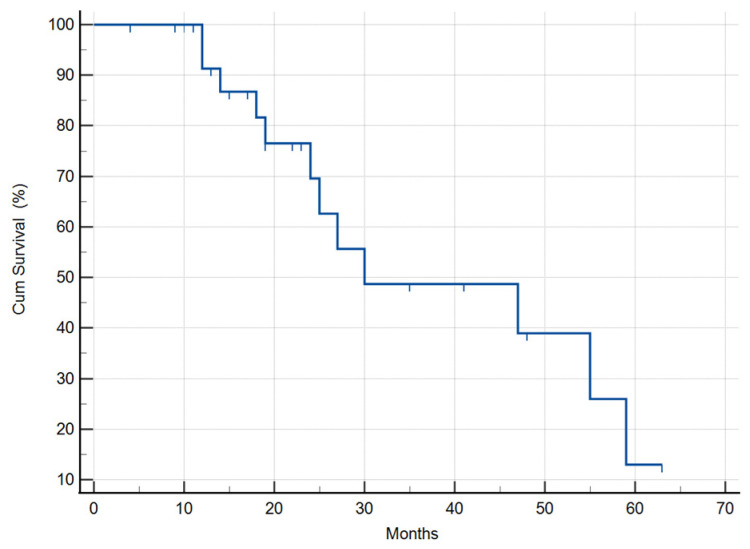
Kaplan–Meier graphs of survival in the study population of the selected 31 GBM patients revealed a 30 (95% CI: 0.5–59.4) months MST, compared to the MST of 12.1–14.4 months based on the historical data of GBM patients.

**Figure 5 biomedicines-11-01989-f005:**
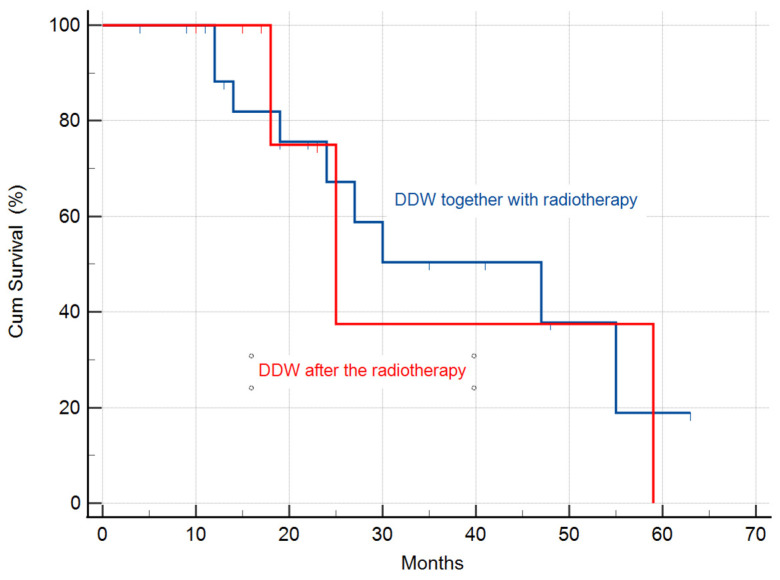
Kaplan–Meier graphs of survival in patients treated with DDW and radiotherapy. The MST was 47 months (95% CI: 18.8–75.1) in the subgroup of patients who started DDW treatment together with radiotherapy, and 25 months (95% CI: 14.5–35.4) for the group where DDW treatment started several months after the radiotherapy was completed.

**Table 1 biomedicines-11-01989-t001:** Results of all GBM patients.

Group	Subgroups	Group Size	MST (Months)
Historical control	Treated with radiotherapy		12.1
	Treated with radiotherapy + TMZ		14.6
All 55 patients		55	30 (95% CI: 9.4–50.5)
	Treated without TMZ	38	27 (95% CI: 18.8–35.1)
	Treated with TMZ	17	42 (95% CI: 14.6–69.3)
	Males	33	25 (95% CI: 15.9–34.0)
	Females	22	42 (95% CI: 18.3–65.6)
Selected 31 patients			30 (95% CI: 0.5–59.4)
	In relapse at the beginning of DDW treatment	20	30 (95% CI: 22.6–37.3)
	In remission at the beginning of DDW treatment	10	47 (95% CI: 0.0–0.0)
	DDW along with radiotherapy	24	47 (95% CI: 18.8–75.1)
	DDW after radiotherapy	7	25 (95% CI: 14.5–35.4)

## Data Availability

No new data were created.

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
