# Peer review of "A Preliminary Study Indicating Improvement in the Median Survival Time of Glioblastoma Multiforme Patients by the Application of Deuterium Depletion in Combination with Conventional Therapy"

_biomedicines, 2023, doi:10.3390/biomedicines11071989_

Round 1

Reviewer 1 Report

Dear Authors, dear Editor,

Draft “biomedicines-2482567-peer-review-v1 - A preliminary study indicating improvement…“ reports a retrospective survival analysis of a group of Hungarian patients with brain cancer (resistant glioblastoma), the short expected survival of whom could be delayed by having them drink industrially produced Deuterium-Depleted Water (DDW) for extended periods.

This is one example of rationally based, yet still “alternative” (this is incorrect, however: the treatment complements standard chemo and radiotherapy) treatment of otherwise unavoidably disrupting, painful and ultimately deadly disease. The Authors’ records, incomplete as they are (I would expect that they were able to have body water Deuterium content assessed), should be published and possibly their article highlighted for reading by the Journal.

I would suggest that an Excel file with pertinent information be included as Supplementary Information, to allow other interested researchers to continue investigation beyond the descriptive that the Authors performed.

Given the fair Authors’ description of their elaboration of the data, I see no flaws and consider this report publishable “as it is”.

Kind regards

Author Response

Response to Reviewer 1.

Thank you for your time evaluating our manuscript and suggesting changes.

As you suggested, Supplementary information will be uploaded with the modified version of the manuscript.

A retrospective study does not allow any invasive interventions, blood collection, to determine the D concentration of the patients. Reference 24 contains that type of information. The study showed a strong correlation between the D concentration of the consumed DDW and the D concentration of body water.

Reviewer 2 Report

The overall goal of the manuscript was to show that deuterium-depleted water (DDW) consumed by patients with GBMs enhanced their standard of care treatment. Patients who consumed DDW during treatment had a medium survival time (MST) greater than patients who did not consume DDW with treatment. In addition, within the DDW-treated group, female patients had a greater MST than their male counterparts. The DDW-treated patients, when administered with TMZ, also had a greater MST than DDW-treated patients who had not received TMZ. Although there was a difference within the DDW-patient groups, it was not statistically significant for both the former and the latter. The authors specified a subgroup of 31 patients who followed the correct procedures for consuming DDW, revealing a greater MST for both a GBM relapse prior to and in remission at the initiation of DDW administration. Moreover, longer MSTs were found with patients consuming DDW simultaneously during radiation treatment compared to at least eight weeks after radiotherapy. Together, these overall survival times were longer than the historical clinical data of primary GBM patients.

In Summary, DDW administration was recommended. The manuscript is original for GBM cancers and relevant for enhancing the treatment of GBMs. The overall gap that the manuscript fills is that DDW administration could be used with standard-of-care treatment to enhance the patient's medium survival. The manuscript contributes to the GBM field and aligns with other publications that deployed DDW for other cancers. The overall caveat is that paper relies on historical patient control data rather than controls initiated along with the experimental groups. Because the manuscript is based on clinical trials, having concurred controls would be a considerable cost. In addition, the historical control data represents a large, well-established dataset. The author's conclusions are consistent with their data. Data and figures are precise, with no changes needed. 

Minor points

1. The authors should discuss deuterium isotope effects regarding the overall level at which deuterium slows down biological reactions. 

2. The authors should also add data on patients who were administered DDW before radiotherapy, if available.

Author Response

Response to Rewiever 2.

Thank you for your time evaluating our manuscript and suggesting changes.

Minor point 1.

We have added a new paragraph with a new citation to the manuscript about the isotope effect, emphasizing the differences between the two hydrogen isotopes.

Minor point 2.

According to the protocol, patients receive radiotherapy within a short time after diagnosis, so there is no possibility to treat patients with DDW for several months before radiotherapy starts.
